Combination of circulating miR-145-5p/miR-191-5p as biomarker for breast cancer detection

Ashirbekov Yeldar eldarasher@mail.ru 1
Abaildayev Arman 1
Omarbayeva Nazgul 2
Botbayev Dauren 1
Belkozhayev Ayaz 1
Askandirova Anel 2
Neupokoyeva Alena 3
Utegenova Gulzhakhan 4
Sharipov Kamalidin 1
Aitkhozhina Nagima 1
1 M. Aitkhozhin Institute of Molecular Biology and Biochemistry , Almaty , Kazakhstan
2 Kazakh Research Institute of Oncology and Radiology , Almaty , Kazakhstan
3 Almaty Branch of National Center for Biotechnology , Almaty , Kazakhstan
4 South Kazakhstan State Pedagogical University , Shymkent , Kazakhstan
Bartolini Barbara
Electronic publication date: 2020 Dec 16
Publication date: 2020
Volume: 8
Electronic Location ID: e10494
Received 2020 Jun 19; Accepted 2020 Nov 13
Copyright: ©2020 Ashirbekov et al.
Copyright year: 2020
Copyright holder: Ashirbekov et al.
License: This is an open access article distributed under the terms of the Creative Commons Attribution License, which permits unrestricted use, distribution, reproduction and adaptation in any medium and for any purpose provided that it is properly attributed. For attribution, the original author(s), title, publication source (PeerJ) and either DOI or URL of the article must be cited.
License URL: https://creativecommons.org/licenses/by/4.0/

Keywords: Breast cancer, Circulating miRNA, Biomarker, Plasma, miR-145, miR-191, miR-21, Diagnosis, Kazakh population

Funding: Science Committee of the Ministry of Education and Science of the Republic of Kazakhstan AP05132207 This research was funded by the Science Committee of the Ministry of Education and Science of the Republic of Kazakhstan (Grant No. AP05132207). The funders had no role in study design, data collection and analysis, decision to publish, or preparation of the manuscript.

==============================
Background

Breast cancer (BC) is the most common cancer among women worldwide. At present, there is a need to search for new, accurate, reliable, minimally invasive and cheap biomarkers in addition to existing methods for the diagnosis and prognosis of BC. The main goal of this study was to test the diagnostic value of six circulating miRNAs in Kazakh women.

Materials and methods

TaqMan-based miRNA profiling was conducted using plasma specimens from 35 BC women patients and 33 healthy women samples (control group).

Results

The level of all seven miRNAs (including endogenous control) normalized by synthetic cel-miR-39 were significantly elevated in the group of BC patients. Normalization using miR-222-3p as endogenous control reduced differences in level of miRNAs between groups; as a result, only three miRNAs were significantly upregulated in the group of BC patients—miR-145-5p (P = 6.5e−12), miR-191-5p (P = 3.7e−10) and miR-21-5p (P = 0.0034). Moreover, ROC analysis showed that the use of miR-145-5p and miR-191-5p, both individually (AUC = 0.931 and 0.904, respectively) or in combination (AUC = 0.984), allows to accurately differentiate BC patients from healthy individuals.

Conclusions

Two plasma miRNAs—miR-145-5p and miR-191-5p—are potential biomarkers for diagnosis of BC in the Kazakh population. The findings need to be further substantiated using a more representative sample.

Introduction

Breast cancer (BC) is the most commonly diagnosed cancer type in women around the world. Just like most cancers, early BC is asymptomatic. This has resulted in late detection of the disease, at which point no therapy is very effective (Höfelmann, Anjos & Ayala, 2014). Mammographic screening of women, in the age range the most at risk to breast cancer, did make the tumor detection at early stages more common and therefore, caused significant reduction in mortality (Onega et al., 2016; Wang, 2017). However, mammography shows a significant number of false positives in women with dense breasts, especially at a younger age. In this regard, mammography screening is confidently recommended for women over 50 years old, although women aged 40–50 years are also at risk of BC (McDonald et al., 2016; Nelson et al., 2016; Phi et al., 2018). Various molecular subtypes of BC that require different therapy (EBCTCG, 2015; Guerrero-Zotano & Arteaga, 2017; Lee & Seo, 2018), individual patient susceptibility to drugs and side effects from drugs (Potosky et al., 2015; Greenlee et al., 2017; Moo et al., 2018) and the development of drug resistance (Li et al., 2020; Zhong et al., 2020) make treatment of this disease more difficult and complicated. The listed difficulties indicate the need for study of new biomarkers that can help in the early detection, diagnosis and prognosis of BC.

Nowadays miRNAs are promising markers for early diagnosis and prognosis of tumors. miRNAs are a large class of small non-coding RNAs that function as negative regulators of most genes in the genome and are involved in important biological processes, such as development, differentiation, apoptosis, proliferation, etc. (Jansson & Lund, 2012). Many studies have highlighted differential expression of certain miRNAs in several cancer types, including BC (Acunzo et al., 2015; Aggarwal, Priyanka & Tuli, 2020).

The property of miRNAs that they can be detected in both tumor cells and biological fluids (in a cell-free form) serves as a major advantage for using these molecules over other oncogenic biomarkers. miRNAs directly enter the bloodstream from primary or metastatic tumors by active secretion, apoptosis or necrosis, and thus changes in the amount of circulating miRNAs can reflect the pathological process (Schwarzenbach, 2017; Sun et al., 2018). In this regard, the level of miRNA-marker can be determined in a minimally invasive way. High stability of miRNA in biological fluids also makes them a very suitable choice as cancer biomarkers (Grasedieck et al., 2012; Glinge et al., 2017). Several miRNAs have been revealed to contribute to the pathological mechanisms of BC progression and many of them have been recommended by previous research studies as diagnostic or prognostic markers (McGuire, Brown & Kerin, 2015; Stückrath et al., 2015; Zhang et al., 2015; Schwarzenbach, 2017; Hamam et al., 2017; Nassar, Nasr & Talhouk, 2017; Shao et al., 2019). The main limitation of currently existing serum biomarkers, including the best of them CA15-3 and CEA, as a marker of BC is the lack of sensitivity for patients with early disease (Duffy, Evoy & McDermott, 2010); miRNA-markers seem to have no such limitations (Schwarzenbach, 2017). It is known that there are some ethnic differences in the pathogenesis of breast cancer (Nakshatri, Anjanappa & Bhat-Nakshatri, 2015; Özdemir & Dotto, 2017; Wu et al., 2020), which is also true for the applicability of miRNAs as markers of BC (Zhao et al., 2010; Nassar et al., 2017; Wu et al., 2020). For this reason, miRNA-markers need to be validated for specific ethnic groups.

The aim of our study was to test the diagnostic value of six circulating miRNAs recommended previously as plasma/serum markers of BC: miR-145-5p (Ng et al., 2013), miR-21-5p (Adhami et al., 2018), miR-210-3p (Jung et al., 2012), miR-29c-3p (Zhang et al., 2015), miR-16-5p (Usmani et al., 2017) and miR-191-5p (Mar-Aguilar et al., 2013) among Kazakh women. To do so, we compared plasma levels of the miRNAs between age-matched BC patients (n = 35) and healthy women (n = 33) from Almaty and Almaty region in Kazakhstan.

Materials & Methods

Subjects

Venous blood of 35 Kazakh women with primary BC was collected at the Kazakh Research Institute of Oncology and Radiology, Almaty, Kazakhstan before therapy in 2019. All patients analyzed had histologic proven BC. The average age of patients was 52.6 ± 11.66. Venous blood of 33 healthy Kazakh women was collected in the Karasai central district hospital in the Almaty region, Kazakhstan in 2019. All controls underwent mammography and were over 40 years old. The average age of the control group was 53.0 ± 7.61. Clinicopathological characteristics of BC patients and control group are presented in Table 1. The study was carried out in compliance with the principles of the Helsinki Declaration, and approved by the local ethics committee of the M. Aitkhozhin Institute of Molecular Biology and Biochemistry, Almaty, Kazakhstan (approval number 185/01-02). All participants provided written informed consent for the use of biomaterials in this study.

Table 1 Clinicopathological characteristics of BC patients group and control group.

Characteristics	BC patients group	Control group	
ER-/ER+	8/27	–	
PR-/PR+	10/25	–	
HER2-/HER2+	27/7	–	
Tumor size: T1/T2/T3/T4	4/28/2/1	–	
Lymph node: Nx/N0/N1-3	5/23/7	–	
Metastases: no/yes	34/1	–	
Ki-67: <20%/≥20%	17/18	–	
Tumor grade: G1/G2/G3	1/28/6	–	
Menarche age: early (≤14)/late (>14)	27/8	16/17	
Menopausal status: pre-/post	14/21	14/19	
Age of first birth: ≤22/≥23	17/16	18/15	
Number of children: 0/≤2/≥3	2/13/20	0/13/20	
Number of unsuccessful pregnancies: 0/1/≥2	11/10/14	8/9/16	
Family history of cancer: no/yes	27/8	24/9	
Alcohol consumption: no/yes	31/4	26/7	

Plasma preparation

Blood was collected in vacuum tubes with sodium citrate, which showed considerable miRNA yield in preliminary tests. Blood was stored at 4 °C and plasma was obtained within 8 h after blood sampling. To obtain plasma, the blood was centrifuged at 1,000 g for 15 min at 4 °C; the upper aqueous phase was transferred to a fresh tube and centrifuged at 2,500 g for 15 min at 4 °C. The resulting plasma was divided into aliquots and stored at −70 °C until the isolation of miRNA step. Before being examined, the plasma was subjected to one freeze-thaw cycle.

Isolation of RNA

Isolation of total RNA from 200 µl of plasma was performed utilizing technique previously developed by Zununi Vahed et al., (2016) with minor modifications. Briefly, deproteinization was carried out according to the standard Trizol method. Then, to precipitate RNA, an equal volume of 2.5M lithium chloride and two volumes of cold ethanol were added and incubated overnight at −70 °C, then centrifuged for 16,000 g for 20 min at 4 °C. The pellets was dried and dissolved in 50 µL of DEPC water, incubating for 5 min at 65 °C. At the stage of Trizol treatment, 20 fmol of synthetic cel-miR-39 was added to the sample. The resulting total RNA sample was stored at −70 °C until use.

Obtaining cDNA and quantitative PCR

Reverse transcription and quantitative PCR was performed using primers and probes from TaqMan MicroRNA Assay (Applied Biosystems, USA). cDNA was obtained using TaqMan MicroRNA Reverse Transcription Kit reagents (Applied Biosystems) according to the manufacturer’s protocol. Quantitative PCR was performed in triplicates using TaqMan Universal Master Mix II with UNG reagents (Applied Biosystems) under the conditions recommended by the manufacturer on the StepOnePlus Real-Time PCR System (Applied Biosystems). Quantitative data was normalized to the level of exogenous spike-in control cel-miR-39 and endogenous control miR-222-3p.

Statistical analysis

Primary processing of the results was carried out in StepOne Software and ExpressionSuite Software. The suitability of endogenous control was evaluated using the NormFinder (Andersen, Jensen & Orntoft, 2004) and GeNorm (Vandesompele et al., 2002) programs. Relative quantification is carried out using the comparative Ct (ΔΔCt) method with modifications as described in the paper (Königshoff et al., 2009). Relative transcript abundance is expressed in ΔCt values (ΔCt = Ctreference − Cttarget). ΔΔCt value (ΔΔCt = ΔCtBC − ΔCtcontrol) was considered as log2 fold change.

Statistics were performed in the Jamovi program (https://www.jamovi.org). Statistical significance of the differences in ΔCt between the groups was calculated using the two-tailed Mann–Whitney U test. P <0.05 was considered statistically significant. Due to the explorative nature of the study no adjustment for multiple testing was performed. The characteristics of the markers were evaluated by ROC analysis using the web-tool easyROC (Goksuluk et al., 2016), and Jamovi. Youden’s index method was used to calculate optimal cut-off points.

Results

Endogenous control selection

To select the best endogenous control, we evaluated the concentration stability of analyzed miRNAs in our sample with the help of NormFinder and GeNorm programs. According to NormFinder, the three best (the lowest) stability values were shown for miR-21-5p, miR-222-3p and miR-29c-3p (Fig. 1A). According to GeNorm, miR-222-3p and miR-29c-3p are the best internal controls for our sample (Fig. 1B). Thus, there are two miRNAs on the overlap of the results of two programs: miR-222-3p and miR-29c-3p.

Figure 1 Selection of endogenous control.

(A) Results from NormFinder: intergroup (bars) and intragroup (whiskers) variation plot and stability value (SV), calculated on their basis; (B) average expression stability values of remaining control candidates during stepwise exclusion of the least stable control candidate, obtained from GeNorm.

Unlike NormFinder, GeNorm does not recommend using miR-21-5p. Also, although NormFinder showed the best stability value for miR-21-5p, intragroup variation in the BC patient group was the largest. This may indicate the heterogeneity of the group and does not exclude the existence of an association between circulating miR-21-5p concentration and some clinicopathological parameter. These considerations, as well as the fact that circulating miR-21-5p has most often been found to be dysregulated in BC (Schwarzenbach, 2017; Adhami et al., 2018), prompted us to abandon it as an endogenous control.

One of the important criteria when choosing endogenous control is their relative abundance. It seems to us that miR-29c-3p is not abundant enough for this role (Ct mean 34.6). Taking into account all the mentioned above, we decided to use miR-222-3p as single endogenous control for our study.

The level of miRNA in the plasma of BC patients in comparison with the control group

The Ct values of the analyzed miRNAs in two groups relative to the spike-in control cel-miR-39 level are shown in Fig. 2A. The concentration of all miRNAs, including miR-222-3p (used later as endogenous control), was significantly elevated in the plasma of BC patients compared to healthy controls. Log2 fold changes higher than one are obtained for miR-145-5p (2.36), miR-191-5p (1.87) and miR-21-5p (1.35) (Table 2).

Figure 2 Differences in ΔCt between BC patients and control group.

(A) Data are normalized to the spike-in control cel-miR-39; (B) data are normalized to the endogenous control miR-222-3p.

Table 2 Cycle threshold values (Ct) and comparative statistics of studied miRNAs between the BC patients group and control group.

miRNA	BCCt mean ± SD	Control Ct mean ± SD	Cel-miR-39 normalization	miR-222-3p normalization	
			BC ΔCt mean ± SE	Control ΔCt mean ± SE	ΔΔCt (95% CI), log2 fold change	P value	BC ΔCt mean ± SE	Control ΔCt mean ± SE	ΔΔCt (95% CI), log2 fold change	P value	
miR-145-5p	29.52 ± 1.52	32.41 ± 1.19	−10.94 ± 0.21	−13.30 ± 0.16	2.36 (1.84; 2.88)	9.2e−10	−0.84 ± 0.11	−2.22 ± 0.12	1.38 (1.06; 1.72)	6.5e−12	
miR-16-5p	22.72 ± 1.70	23.94 ± 1.46	−4.14 ± 0.20	−4.83 ± 0.18	0.69 (0.15; 1.23)	0.020	5.96 ± 0.15	6.25 ± 0.14	−0.29(−0.69; 0.12)	0.081	
miR-191-5p	26.76 ± 1.76	29.17 ± 1.17	−8.18 ± 0.26	−10.05 ± 0.14	1.87 (1.28; 2.46)	2.3e−08	1.92 ± 0.08	1.02 ± 0.09	0.89 (0.66; 1.13)	3.7e−10	
miR-21-5p	26.24 ± 1.84	28.13 ± 1.10	−7.66 ± 0.18	−9.02 ± 0.14	1.35 (0.89; 1.82)	3.2e−07	2.44 ± 1.17	2.06 ± 0.09	0.38 (−0.04; 0.76)	0.0034	
miR-210-3p	32.31 ± 1.38	33.53 ± 1.09	−13.73 ± 0.17	−14.42 ± 0.15	0.69 (0.24; 1.14)	0.0023	−3.63 ± 0.15	−3.35 ± 0.11	−0.29(−0.65; 0.08)	0.100	
miR-29c-3p	33.87 ± 1.81	35.38 ± 0.96	−15.29 ± 0.22	−16.27 ± 0.10	0.98 (0.49; 1.47)	0.0002	−5.19 ± 0.10	−5.19 ± 0.07	0.01 (−0.24; 0.25)	0.845	
miR-222-3p	28.68 ± 1.52	30.19 ± 0.84	−10.10 ± 0.20	−11.08 ± 0.08	0.98 (0.53; 1.42)	0.0006	–	–	–	–	
cel-miR-39	18.58 ± 1.10	19.11 ± 0.67	–	–	–	–	−10.10 ± 0.20	−11.08 ± 0.08	−0.98 (−1.42; −0.53)	0.0002	

When quantitative data were normalized to miR-222-5p, the levels of miR-145-5p, miR-191-5p and miR-21-5p in the BC group were significantly increased compared to healthy controls (Fig. 2B). Differences between groups in miR-16-5p, miR-210-3p, and miR-29c-3p concentrations were not significant. Compared to cel-miR-39 normalization, log2 fold change significantly decreased: only one miRNA exceeded one—miR-145-5p (1.38). Relative to the endogenous control, the level of cel-miR-39 was significantly lower in the group of BC patients (ΔΔCt =  − 0.98, P = 0.0004) with a wider range of ΔCt values compared to the control group.

Associations with clinicopathological parameters

The results of comparisons between groups with different clinicopathological characteristics are presented in Table 3. When normalized to endogenous control miR-222-3p, the level of miR-145-5p was significantly higher (P = 0.043) and the level of miR-191-5p was significantly lower (P = 0.006) in patients with HER2 positive tumor compared to patients with HER2 negative tumor. The level of miR-21-5p in patients with high Ki-67 (≥20%) was significantly higher compared to patients with low Ki-67 (P = 0.003). The level of miR-210-3p and miR-145-5p in patients with poorly differentiated tumor (grade G3) were significantly higher compared to patients with moderately differentiated tumor (grade G2) (P = 0.007 and 0.033, respectively). In the group of BC patients, levels of miR-145-5p and miR-21-5p were significantly higher in women with early menarche compared to women with late menarche (P = 0.009 and 0.022, respectively). In the control group, the level of miR-21-5p in women with two or less children was significantly higher compared to women with more than two children (P = 0.011). In the control group, the level of miR-29c-3p in women over 50 years old was significantly lower compared to women younger than or 50 years old (P = 0.008). In the control group, the level of miR-191-5p in women with a positive family history of cancer was significantly lower compared to women without it (P = 0.029). Differences in the level of the analyzed miRNAs between the groups, categorized by other clinicopathological parameters were not significant.

Table 3 P values for ΔCt comparisons between groups with different clinicopathological characteristics, after normalization to miR-222-3p.

Clinicopathological characteristics	miR-145-5p	miR-16-5p	miR-191-5p	miR-21-5p	miR-210-3p	miR-29c-3p	
BC patients group							
ER- vs ER+	0.630	0.714	0.862	0.269	0.832	0.428	
PR- vs PR+	0.287	0.627	0.553	0.339	0.577	0.122	
HER2- vs HER2+	0.043	0.559	0.006	0.379	0.191	0.771	
N0 vs N1-3	0.086	0.441	0.190	0.246	0.810	0.360	
Ki-67 <20% vs ≥20%	0.096	0.134	0.089	0.003	0.708	0.405	
Tumor grade: G2 vs G3	0.033	0.066	0.644	0.297	0.007	0.676	
Age: <50 vs. ≥50	0.257	0.987	0.906	0.371	0.191	0.749	
Menarche age: ≤14 vs >14	0.009	0.743	0.166	0.022	0.802	0.862	
Menopausal status: pre- vs post	0.096	0.908	0.517	0.249	0.118	0.881	
Age of first birth: ≤22 vs >22	0.063	0.683	0.345	0.873	0.102	0.276	
Number of children: ≤2 vs >2	0.987	0.347	0.139	0.107	0.521	0.099	
Unsuccessful pregnancies: 0 vs >0	0.316	0.061	0.713	0.163	0.300	0.099	
Family history of cancer: no vs yes	0.576	0.499	0.550	0.143	0.286	0.143	
Alcohol consumption: no vs yes	0.093	0.378	0.233	0.745	0.379	0.565	
Control group							
Age: <50 vs. ≥50	0.501	0.986	0.102	0.842	0.137	0.008	
Menarche age: ≤14 vs >14	0.402	0.118	0.087	0.581	0.276	0.402	
Menopausal status: pre- vs post-	0.240	0.186	0.872	0.553	0.815	0.114	
Age of first birth: ≤22 vs >22	0.486	0.929	0.166	0.442	0.401	0.901	
Number of children: ≤2 vs >2	0.478	0.316	0.392	0.011	0.235	0.730	
Unsuccessful pregnancies: 0 vs >0	0.696	0.272	0.886	0.067	0.127	0.726	
Family history of cancer: no vs yes	0.796	0.438	0.029	0.592	0.179	0.564	
Alcohol consumption: no vs yes	0.352	0.215	0.780	0.682	0.352	0.249	
Note.

p values <0.05 are in bold.

We also found statistically significant differences in the distribution of women with early and late menarche between BC and control groups (P = 0.023, OR = 3.59, 95% CI [1.26–10.18]), and an inverse correlation between the level of Ki-67 and the age of BC patients (Spearman’s rho = −0.507, P = 0.0019).

We did not consider differences between groups divided by clinicopathological parameters based on data normalized to spike-in cel-miR-39, due to doubtful results (see Discussion for details).

ROC analysis

To test the ability of our miRNAs to distinguish BC patients from healthy individuals, we performed a ROC analysis, the results are presented in Table 4. When normalized to cel-miR-39, the largest area under the ROC curve (AUC) was obtained for miR-145-5p (0.932); miR-191-5p and miR-21-5p were far behind with values close to each other (0.868 and 0.842, respectively) (Fig. 3A). AUC for the remaining 4 miRNAs was lower than 0.8 (Fig. 3B). Using combination models of the three best markers did not increase at least a hundredth of the best individual AUC.

Table 4 ROC analysis results for potential markers and their combinations.

Classes	Potential markers/ combinations	Cel-miR-39 normalization	miR-222-3p normalization	
		AUC	Optimal cut-of value (point)	Specificity	Sensitivity	Accuracy	AUC	Optimal cut-of value (point)	Specificity	Sensitivity	Accuracy	
Controls vs BC patients	miR-145-5p	0.932	0.535 (−12.17)	0.857	0.909	0.882	0.932	0.32 (−1.77)	0.788	0.971	0.882	
	miR-16-5p	0.664	0.51 (−4.45)	0.606	0.686	0.647	–	–	–	–	–	
	miR-191-5p	0.868	0.60 (−9.00)	0.939	0.714	0.824	0.904	0.421 (1.37)	0.818	0.914	0.868	
	miR-21-5p	0.842	0.58 (−8.21)	0.848	0.743	0.794	0.705	0.54 (2.42)	0.848	0.714	0.779	
	miR-210-3p	0.713	0.55 (−13.86)	0.758	0.657	0.706	–	–	–	–	–	
	miR-222-3p	0.760	0.549 (−10.57)	0.879	0.657	0.765	–	–	–	–	–	
	miR-29c-3p	0.739	0.68 (−15.23)	0.970	0.457	0.706	–	–	–	–	–	
	miR-145-5p + miR-191-5p	0.930	0.52	0.879	0.886	0.882	0.984	0.72	1.000	0.943	0.971	
	miR-145-5p + miR-21-5p	0.936	0.44	0.818	0.943	0.882	0.932	0.44	0.818	0.914	0.868	
	miR-191-5p + miR-21-5p	0.875	0.36	0.697	0.914	0.809	0.919	0.53	0.879	0.857	0.868	
	miR-145-5p + miR-191-5p + miR-21-5p	0.933	0.605	0.939	0.829	0.882	0.984	0.72	1.000	0.943	0.971	
HER2- vs HER2+	miR-145-5p	–	–	–	–	–	0.751	0.15 (−0.98)	0.481	1.000	0.588	
	miR-191-5p	–	–	–	–	–	0.831	0.147 (1.98)	0.667	1.000	0.735	
Ki-67: <20% vs ≥20%	miR-21-5p	–	–	–	–	–	0.791	0.506 (2.55)	0.706	0.944	0.829	
Tumor grade: G2 vs G3	miR-145-5p	–	–	–	–	–	0.780	0.32 (−0.25)	0.667	0.964	0.912	
	miR-210-3p	–	–	–	–	–	0.845	0.10 (−3.73)	1.000	0.679	0.735	

Figure 3 ROC plots for miRNAs, showing significant differences in plasma levels between BC patients group and control group.

(A) miR-145-5p, miR-191-5p and miR-21-5p, normalized to cel-miR-39; (B) miR-16-5p, miR-210-3p, miR-222-3p and miR-29c-3p, normalized to cel-miR-39; (C) miR-145-5p, miR-191-5p and miR-21-5p, normalized to miR-222-3p; (D) Combination of miR-145-5p and miR-191-5p, normalized to miR-222-3p.

When normalized to miR-222-3p, only three miRNAs, that showed significant differences in concentration between BC patients and controls, were tested for suitability as diagnostic markers. Although log2 fold change was significantly reduced relative to cel-miR-39 normalization, the AUC for miR-145-5p was the same 0.932, and for miR-191-5p even increased and amounted to 0.904 (Fig. 3C). The diagnostic effectiveness of miR-21-5p significantly decreased to AUC = 0.705. The combination of miR-145-5p and miR-191-5p in one model made it possible to increase AUC to 0.984 (Fig. 3D) with the highest specificity, good sensitivity (0.943) and accuracy of separation (97%). The addition of miR-21-5p to this combination did not lead to changes in indicators.

We also tested the ability of miRNAs to separate BC patients according to clinicopathological parameters. ROC analysis showed that using miR-145-5p and miR-191-5p it was possible to distinguish patients with HER2 negative tumors from patients with HER2 positive tumors with 58% and 74% accuracy, respectively; using miR-21-5p it was possible to divide patients into low and high Ki-67 groups (<20% vs ≥20%) with 83% accuracy; using miR-145-5p and miR-210-3p it was possible to distinguish patients with moderately differentiated and poorly differentiated tumors with 92% and 74% accuracy, respectively.

Discussion

When planning the study, in accordance with literature, we selected 5 miRNAs as candidate markers (miR-21-5p, miR-145-5p, miR-210-3p, miR-222-3p and miR-29c-3p), one miRNA as candidate marker or endogenous control (miR-16-5p), one miRNA as endogenous control (miR-191-5p), and one miRNA as exogenous spike-in control (cel-miR-39). However, for the reasons stated below, we decided not to use the spike-in control and chose miR-222-3p as the endogenous control.

When working with bio-fluids, the amount of input biomaterial is easily standardized by the specified volume of the sample, thereby it is possible to take into account the differences that arise during RNA isolation. This is achieved by adding to the sample a certain dose of synthetic miRNA at the step of lysis (Kroh et al., 2010). The lack of reliable and universally accepted endogenous control for miRNA data normalization (Schwarzenbach et al., 2015) determines the relevance of using a spike-in control. Therefore, we first tested spike-in control normalization method.

When we used cel-miR-39 as reference, the average ΔCt values for all 7 miRNAs in BC patients were significantly higher than in controls. These results seem suspicious, although it is possible that they reflect the actual difference between compared groups. Second explanation: blood specimens of the compared groups differed in the degree of hemolysis, although plasma with visually distinct hemolysis was excluded from the analysis in advance. However, Appierto et al. showed that the initial stages of hemolysis are visually indistinguishable (Appierto et al., 2014). In our case, the level of miR-16-5p, which is considered as a marker of hemolysis (Pizzamiglio et al., 2017), varied less in comparison with other miRNAs. The third explanation: two groups differed in the content of plasma proteins and lipids associated with miRNA, which may affect the efficiency of miRNA isolation, as suggested by Sourvinou, Markou & Lianidou (2013). They found that the Trizol method yielded a reduced amount of spike-in cel-miR-39 compared to endogenous miR-21. In our case, the average Ct value for cel-miR-39 in the group of BC patients was significantly lower than that in the control group (P = 0.003), but for targeted miRNAs the difference was even more considerable. The obtained data indicate better efficiency of RNA isolation in the group of BC patients, but it is unclear whether the yield of the added synthetic cel-miR-39 and endogenous miRNA in each of the two groups is equal. Due to the ambiguity in this matter, we could not confidently use the spike-in control to normalize our data. Perhaps using column-based RNA isolation methods would solve this problem, as shown by Sourvinou, Markou & Lianidou (2013).

Since the spike-in control was inappropriate, we evaluated the concentration stability of endogenous miRNAs to determine its suitability as an internal control. Surprisingly, both initial candidates for reference, miR-191-5p and miR-16-5p, were inferior in stability to other miRNAs. Based on an analysis of concentration stability of our miRNA, and also taking into account the relative abundance of transcripts, we chose miR-222-3p as reference, although initially we selected it as target miRNA for the study in accordance with literature screening (Hu et al., 2012; Song et al., 2017; Kim et al., 2019). Previously, this miRNA was already used as a reference in such studies (Tay et al., 2017). After replacing spike-in cel-miR-39 by endogenous miR-222-3p the difference in the target miRNAs level between the two groups considerably decreased, and as a result, the number of dysregulated miRNAs was reduced to three. Despite this, according to the ROC analysis, the ability of miR-145-5p to distinguish BC patients from controls remained the same; for miR-191-5p it even increased; and the combination of the two made it possible to further improve the separation efficiency. In addition, based on these data, we found associations with clinicopathological parameters for some miRNAs. These arguments suggest that we selected the endogenous control correctly, and our results reflect the real state of things.

miR-191-5p is probably the most commonly used as endogenous control in quantitative studies of circulating miRNAs. To date, there is evidence of important role of miR-191 in tumorigenesis and its dysregulation in a wide range of cancers, including BC (Gao et al., 2017; Zhang et al., 2018). Two studies showed the association of circulating miR-191 with BC (Ng et al., 2013; Mar-Aguilar et al., 2013). In agreement with these data, we also found a significant upregulation of circulating miR-191-5p in BC patients compared to healthy women. In addition, the concentration of miR-191-5p differed in plasma of BC patients depending on HER-2 status of the tumor.

miR-16-5p has also been frequently used previously as an endogenous control (McDermott, Kerin & Miller, 2013; Donati, Ciuffi & Brandi, 2019). At the same time, several studies report about increased miR-16-5p concentrations in plasma of BC patients compared to healthy controls (Hu et al., 2012; Ng et al., 2013; Stückrath et al., 2015; Usmani et al., 2017). A meta-analysis of the diagnostic and prognostic value of miR-16 showed that its use as a biomarker is more applicable in Asian populations (Cui, 2015). Our data are not consistent with the aforementioned studies: we found no significant differences in plasma levels of miR-16-5p between breast cancer patients and the controls in the Kazakh population.

miR-145-5p showed the most significant association with BC in our study. This miRNA inhibits the expression of certain oncogenes and thus acts as a tumor suppressor (Sachdeva et al., 2009). In accordance with this concept, most previous studies reported about reduced level of circulating miR-145 in BC patients compared to controls (Ng et al., 2013; Kodahl et al., 2014; Hu et al., 2015). In contrast, in the aforementioned study, Mar-Aguilar et al. (2013) found elevated mir-145-5p level in the serum of BC patients, which is consistent with our data. Thus, according to the identified associations of miR-145-5p and miR-191-5p, our Kazakh population is similar to the Mexican one, and differs from other studied populations. Our results in comparison with published data confirm the thesis that the applicability of the miRNA-marker needs to be verified for certain ethnic group. The revealed differences in plasma miR-145-5p concentration between BC patients with early and late menarche may help to further understand the role of this miRNA in the pathogenesis of BC.

The most frequently mentioned circulating miRNA in association with BC is miR-21-5p (Schwarzenbach, 2017; Adhami et al., 2018). We also confirm this association in the Kazakh population. The NormFinder showed a wide range of miR-21-5p variation in the BC patient group, which indicates the heterogeneity of this group. Indeed, we found significant differences in miR-21-5p level between groups separated by some clinicopathological parameters. We found its significantly increased concentration in the plasma of BC patients with high Ki-67, which is consistent with the data that miR-21 promotes BC proliferation (Qiu et al., 2018; Wang et al., 2019). Early menarche and reduced breastfeeding are considered as risk factors for BC (Jeong et al., 2017; Khalis et al., 2018). We found associations of both factors with elevation of miR-21-5p in plasma of Kazakh women. According to our data, miR-21-5p can play an important role in the development of BC in women with these risk factors.

miR-210 is known as a marker of hypoxia during tumor development; and in BC, hypoxia is associated with resistance to therapy and poor prognosis (Camps et al., 2008; Pasculli et al., 2019). Previous studies have shown that dysregulation of circulating miR-210 in BC is associated with tumor presence and lymph node metastasis in patients with HER-2 positive BC (Jung et al., 2012), metastases (Markou et al., 2016; Madhavan et al., 2016) and resistance to chemotherapy (Jung et al., 2012; Shao et al., 2019). In our study, unfortunately, patients with lymph node metastasis were insignificantly represented (N = 7); and there was only one patient with distant metastases. We found no difference in the plasma levels of miR-210-3p in these patients compared to other patients. Instead, we found increased levels of miR-210-3p in patients with poorly differentiated tumor (grade 3) compared with patients with moderate differentiated tumor (grade 2). The findings are consistent with the result of a previous study, which showed an increased expression of miR-210 in poorly differentiated tumors compared to well-differentiated tumors (Wu, 2020). Thus, we have shown that circulating miR-210-3p can be a marker of aggressive, poorly differentiated tumors.

miR-29 has been shown to have an important role in cancer development (Kwon et al., 2018). In most cancer, miR-29 acts as a tumor suppressor by promoting tumor cell apoptosis, by suppressing DNA methylation of tumor-suppressor genes and by reducing proliferation of tumors and by increasing chemosensitivity (Jiang et al., 2014). In contrast, in BC, miR-29 acts as an oncogene by inhibiting fibrosis and thereby promoting epithelial-mesenchymal transition (Jiang et al., 2014; Wang et al., 2017). In line with this, it has been shown that miR-29 is up-regulated both in breast tumors and in the serum of BC patients (Wu et al., 2012; Zhang et al., 2015). But, we found no significant differences in plasma miR-29c-3p concentration between BC patients and controls in the Kazakh population. Instead, we found that level of circulating miR-29c-3p decrease in women (healthy controls) after age 50 compared to younger women. Taking into account the anti-fibrotic activity of miR-29, our data are consistent with the fact that fibrotic processes increase with advancing age (Nho, 2015).

To evaluate the diagnostic effectiveness of potential markers, we performed a ROC analysis. We identified two miRNAs—miR-145-5p and miR-191-5p, which are able to accurately distinguish patients with BC from healthy women, both individually and in combination. The most effective is their combination model, which showed 97% accuracy in the separation of two groups—66 out of 68 women were classified correctly. The applicability of the revealed diagnostic capabilities of miRNAs according to clinicopathological parameters is debatable.

Although we found a promising combination of miRNA-markers to differentiate BC patients from healthy people, there are a few suggestions for further research. As the sample size is small, further validations in large cohort are recommended. The majority of BC patients in our study had T2 tumors; so, it is necessary to check whether the data obtained are valid for other stages of tumor progression. Also, it is desirable to investigate whether our miRNAs are reversed in plasma of BC patients undergoing treatment. In addition, it would be interesting to study the expression of this miRNAs in tumor tissue to test the secretory hypothesis.

Conclusions

When using spike-in cel-miR-39 as a reference, we obtained doubtful results. Some possible reasons are unequal isolation efficiency of endogenous and spike-in miRNA in each of the two groups, visually undetectable hemolysis, or other unknown factors. Endogenous controls selected according to the literature should be verified in the current study. Based on the results of the analysis of concentration stability as well as taking into account the relative abundance of transcripts, we selected miR-222-3p as the endogenous control for our samples.

We revealed three plasma miRNAs (miR-145-5p, miR-191-5p and miR-21-5p) significantly elevated in BC patients compared to control group. ROC analysis showed, that using miR-145-5p and miR-191-5p (both individually and in combination), it is possible to separate BC patients from healthy individuals quite accurately, therefore, these miRNAs should be considered as potential biomarkers for BC detection in Kazakh population. The inconsistency of some of our results with published data suggests that it is necessary to verify biomarkers for certain ethnic group. The findings need to be confirmed on a more representative cohort of samples.

Supplemental Information

Supplemental Information 1 Experimental and clinicopathological data

Click here for additional data file.

Additional Information and Declarations

Competing Interests

Author Contributions

Human Ethics

Data Availability

The authors declare there are no competing interests.

Yeldar Ashirbekov conceived and designed the experiments, analyzed the data, prepared figures and/or tables, authored or reviewed drafts of the paper, and approved the final draft.

Arman Abaildayev conceived and designed the experiments, performed the experiments, analyzed the data, prepared figures and/or tables, authored or reviewed drafts of the paper, and approved the final draft.

Nazgul Omarbayeva conceived and designed the experiments, performed the experiments, authored or reviewed drafts of the paper, collection of biomaterial, and approved the final draft.

Dauren Botbayev and Anel Askandirova performed the experiments, prepared figures and/or tables, collection of biomaterial, and approved the final draft.

Ayaz Belkozhayev performed the experiments, prepared figures and/or tables, and approved the final draft.

Alena Neupokoyeva, Kamalidin Sharipov and Nagima Aitkhozhina conceived and designed the experiments, authored or reviewed drafts of the paper, and approved the final draft.

Gulzhakhan Utegenova analyzed the data, authored or reviewed drafts of the paper, and approved the final draft.

The following information was supplied relating to ethical approvals (i.e., approving body and any reference numbers):

Local ethics committee of the M. Aitkhozhin Institute of Molecular Biology and Biochemistry, Almaty, Kazakhstan (185/01-02).

The following information was supplied regarding data availability:

Experimental and clinico-pathological data is available as Supplemental File.

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
