# Peer review of "Combination of circulating miR-145-5p/miR-191-5p as biomarker for breast cancer detection"

_PeerJ, doi:10.7717/peerj.10494_

## Round 0.1 · original submission · Major Revisions

The manuscript addresses some important facts concerning the use of miRNA as markers for breast cancer detection. Nevertheless, to strengthen the results, it is mandatory to improve some data analysis and better describe the sample collection, where possible, according to reviewers' suggestions.

·

Basic reporting

The article is well written regarding language and data structuring.
Sentence in Line 38 is not clear (until recently? Meaning like 10 years ago).
There isn’t enough literature in the introduction or the discussion to justify the gap in knowledge. The main reason for the need of liquid biopsy biomarker for early BC detection is the decrease in sensitivity of mammography in dense breasts especially in young females who are not recommended to do the test.
Figures require more clarity and labelling regarding acronyms. Figure 1 is hard to follow. What is SV? What does the x and y axis represent in Fig1A?

Experimental design

-unclear gap in knowledge: The choice of microRNA studied is not well explained. In the line 69, it was described that 6 microRNA were investigated in circulation without mentioning them but three of them were later considered as endogenous controls. It is not enough to study only 3 miRNA in circulation without any explanation about the choice of miRNA.
-Unclear results: At the beginning of the article, GeNorm was used to show that the optimal number of control miRNA is three but only one control miRNA was used. Furthermore, when NormFinder was used. why were miR-222-3p and miR-29c-3p as endogenous control? miR-16 and RNU6B are widely used as endogenous controls. In your data, miR-16 was not significantly dysregulated. Which level of stability value that determines whether it is considered as an endogenous control?
-It is said that the normal women were selected by ethnicity and age: were they matched?

Validity of the findings

Data analysis is not sound. When analyzing the real time PCR data, it was concluded that miR-145 and miR-191 were upregulated. We can’t compare delta CT values but either fold change which is 2^(-delta delta CT) or negative delta CT. miR-145 has delta delta CT=2.36 So Fold change= 0.195 meaning it is downregulated in BC compared to control or you can say that negative delta ct of control is higher than negative delta CT of BC. Data are not well analyzed and require modification. For more guidance on analysis, refer to Si et al., 2013 or Nassar et al., 2014.
Regarding the association with clinic-pathological parameter, the results should be represented in a figure or table. Also, it is not clear what was compared between the 2 groups. Levels of miRNA meaning fold change? Delta CT? Reanalyze the data according to what was discussed above.
The ROC curve should be rechecked after reanalyzing the data.
Discussion should be remodified accordingly.

Additional comments

The article requires reanalysis to ensure solid results. The choice of endogenous control could be based on the literature if the cel-miR-39 is not valid. Furthermore, the choice of microRNA should be further discussed. The article focused alot on the choice of endogenous control more than the dysregulated miRNA.

Reviewer 2 ·

Basic reporting

Minor corrections:
Formula for expression stability
Verification of miRNA expression in tumor tissue and cell lines (for secretory hypothesis)
Not discussed about disadvantages of serum biomarkers, such as CA15.3
Was blood collected from patients under treatment?
What does a more representative sample (Line 279) entail?
How many freeze-thaw cycles of the stored plasma?
Line 19 need for new, accurate (comma missing)...
Line 20 The main goal of this study…
Line 22 was conducted using age-matched plasma specimens from 35 BC women patients and 30 healthy women samples (control group).
Line 24 The expression level of…
Line 26 as endogenous control…
Line 28 Moreover, ROC analysis…
Line 30 allows to accurately differentiate BC patients from healthy individuals.
Line 33 These findings need to be further substantiated using a more representative sample.
Line 37 Duplicated line 18. Rewrite please.
Line 37 Just like most cancers, early BC is asymptomatic.
Line 38 This has resulted in late detection of the disease, at which point no therapy is very effective.
Line 40 Mammographic screening of women, in the age range the most at risk to breast cancer, did make the tumor detection at early stages more common and therefore, caused significant reduction in mortality….
Line 43 Various molecular subtypes of BC make treatment of this disease more difficult and complicated.
Line 44 Kindly re-write the sentence for better clarity.
Line 49 miRNAs are a large class of…
Line 51 cellular development
Line 52 Many studies have highlighted differential expression of certain miRNAs in several cancer types, including BC.
Line 55 The property of miRNAs that they can be detected in both tumor cells and biological fluids (in a cell-free form) serves as a major advantage for using these molecules over other oncogenic biomarkers.
Line 56 Micro RNAs directly enter the bloodstream from primary or metastatic tumors by active secretion…
Line 60 High stability of miRNA in biological fluids also makes them a very suitable choice as cancer biomarkers.
Line 62 Several miRNAs have been revealed to contribute to the pathological mechanisms of BC progression and many of them have been recommended by previous research studies as diagnostic or prognostic markers.
Line 69 The aim of our study…
Line 70 To do so, we compared plasma levels of the six selected miRNAs between age-matched breast cancer patients (n= 35) and healthy women (n= 30) from Almaty and Almaty region in Kazakhstan. Please add reference as well.
Line 77 Venous blood of 30 patients…
Line 80 Clinicopathological…
Line 84 biomaterials…
Line 90 until the isolation of miRNA step…
Line 92 performed utilizing technique
Line 96 The pellet was discarded
Line 104 triplicates
Line 107 data was normalized
Line 110 Primary Processing of the results…
Line 119 Youden’s index method was used to calculate optimal cut-off points.
Line 123 analyzed used twice in the same sentence.
Line 127 inter-group
Line 133 According to GeNorm,…. are the best internal controls for…
Line 141 What is the significance of miR210-3p in this context?
Line 166 In the control group, the level of
Line 169 groups, categorized by other
Line 174 State the reason.
Line 180 were far behind with values close to each other
Line 232 ...reference although initially we selected it as target miRNA for the study in accordance with literature screening.
Line 268 We found associations of both factors
Line 275 30 +25 = 65
Line 279 need confirmation on a more representative cohort of samples.
Line 294 Duplicate of Line 279. Please edit.

Experimental design

Although the manuscript sheds light on previously unclear details regarding the miRNAs that are differentially expressed in the plasma of breast cancer patients compared to healthy control in Kazakhstan, it fails to address the following in its experimental design:

1) The authors do not reference the studies or the sources through which they identify the 6 miRNAs (noeither in methods nor in results) that they test as biomarkers for breast cancer in Kazhak women.

2) The study is claimed to be exploratory. However, the qPCR data needs to be validated via in vitro experiments -
a) Testing some of the tumors, derived from patients from whom plasma was collected for the study, in nude mice and conducting similar qPCR studies on mice plasma to verify differential expression of miRNAs.
b) Staining tumors excised from patients to determine whether the miRNAs identified in plasma as biomarkers are secreted by primary tumors or not.

3) Authors present no information whether mammography was used as a paired diagnostic test.

4) Raw data for RNA measurement should be presented, specifically A260/280 ratio, to determine any difference in the quality and purity of RNA extracted from patient and control samples.

Validity of the findings

No additional comments.

---

## Round 0.2 · Minor Revisions

Thanks for addressing the issue previously reported.

I would recommend making the very minor changes suggested by the reviewer to finalize your manuscript.

·

Basic reporting

In line 98, reference the articles that supporting choosing the six miRNA.were they proved in circulation? Give at least one refernce for each miRNA.

Change "Mir" or "MiR" in line 325, 333, 356, 370 to "miR" and even screen figure legends and the whole text for that.

Add info about the healthy controls in the material and method where you specify that they undergone mammography and were older than 40 years old.

Experimental design

No comment

Validity of the findings

No comment

Additional comments

The article has drastically improved.

Reviewer 2 ·

Basic reporting

The authors have incorporated the suggested changes regarding language.

Experimental design

na

Validity of the findings

na

Additional comments

Thank you for incorporating the recommended changes. The manuscript is good to go!

---

## Round 0.3 · accepted · Accept

I would like to thank you and the co-authors for the constant effort profused to adapt your manuscript to the reviewers' indications.